# What else does attention need: Neurosymbolic approaches to general logical reasoning in LLMs?

## Abstract

General logical reasoning is perhaps the most impenetrable challenge for large language models (LLMs). We define general logical reasoning as the ability to reason deductively on domain-agnostic tasks. Current LLMs fail to reason deterministically and are not interpretable. As such, there has been a recent surge in interest in neurosymbolic AI, a research area that attempts to incorporate logic into neural networks. We first identify two main neurosymbolic approaches to improving logical reasoning: (i) the *integrative approach* comprising models where symbolic reasoning is contained within the neural network, and (ii) the *hybrid approach* comprising models where a symbolic solver, separate from the neural network, performs symbolic reasoning. Both contain AI systems with promising results on domain-specific logical reasoning benchmarks. However, their performance on domain-agnostic benchmarks is understudied. To the best of our knowledge, there has not been a comparison of the contrasting approaches that answers the following question: Which approach is more promising for developing general logical reasoning without sacrificing the capabilities of existing LLMs? To analyze their potential, the following best-in-class domain-agnostic models are introduced: Logic Neural Network (LNN), which uses the integrative approach, and LLM-Symbolic Solver (LLM-SS), which uses the hybrid approach. Compared to the current state-of-the-art neurosymbolic models, LNN achieves faster convergence and higher accuracy while LLM-SS delivers a lower error rate. Using both models as case studies and representatives of each approach, our analysis demonstrates that the hybrid approach is more promising for developing general logical reasoning because (i) its reasoning chain is more interpretable than the integrative approach, and (ii) it retains the capabilities and advantages of existing LLMs. To support future works using the hybrid approach to improve general logical reasoning, we propose a generalizable neurosymbolic framework based on LLM-SS that is modular by design, model-agnostic, domain-agnostic, and requires little to no human input.

## 1 Introduction

Following the seminal paper "Attention is all you need" (Vaswani et al., 2017), the emergence of artificial general intelligence (AGI) appears closer than ever. State-of-the-art models, e.g. GPT-4 (Achiam et al., 2023) and Gemini 1.5 (Reid et al., 2024), steadily inch higher and higher on a wide array of benchmarks, from text summarization to code generation (Achiam et al., 2023; Chen et al., 2021). Nonetheless, LLMs continue to exhibit serious deficiencies in their ability to perform logical reasoning (Huang & Chang, 2022). Despite the gradual rise in accuracy in logical reasoning benchmarks, LLMs' fundamental problems have not been mitigated.

Consider the example in Fig. 1, which shows a chain-of-thought (CoT) response to a question from the StrategyQA dataset (Geva et al., 2021), generated by Wei et al. (2022). The text highlighted in yellow shows the LLM's premises, i.e. the evidence used to support the conclusion, while the text highlighted in green shows the LLM's conclusion. This example highlights two primary flaws in LLMs: (1) the premises do not lead to the conclusion. Given the two premises, the final answer should have been true; (2) the conclusion may not have been derived from the premises at all. This problem is more subtle, for we naturally assume

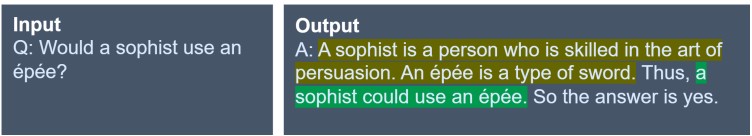

Figure 1: An example CoT output from a question in StrategyQA dataset in Wei et al. (2022)

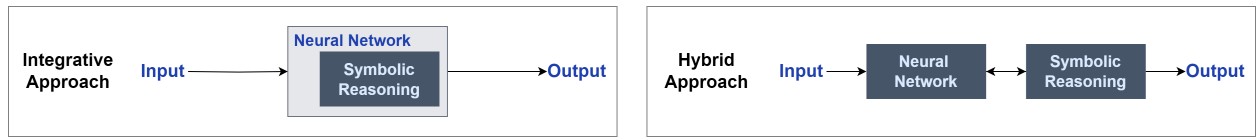

Figure 2: Integrative and hybrid approaches to neurosymbolic AI for general logical reasoning

that if the conclusion does not follow from the premises, it implies the LLM has made a logical reasoning mistake. However, we cannot definitively state that the LLM inferred based on any and all of the premises, which are further explained below.

Both problems are the results of LLM's model architecture. LLMs are, broadly speaking, a combination of linear and non-linear matrix operations, e.g. ReLU (Agarap, 2018) and sigmoid functions (Rasamoelina et al., 2020), sprinkled with neural network techniques, e.g. batch normalization (Ioffe & Szegedy, 2015) and dropout (Srivastava et al., 2014). As a result, they belong to the connectionist approach to AI (Fahlman & Hinton, 1987), and are therefore probabilistic, rather than deterministic, in nature by design. Determinism is defined as a model's ability to produce the same result given the same input regardless of the random seed (if any). Deductive reasoning is, however, deterministic, rather than probabilistic, thus causing Flaw (1). Flaw (2) cannot be ruled out or resolved due to the lack of interpretability of Transformer-based architectures: the use of high-dimensional embeddings and matrix operations obfuscates the underlying premises and ideas. Evidently, so long as LLM architecture remains the same, no amount of parameters or training data will solve these fundamental problems in logical reasoning.

In response to these challenges, neurosymbolic AI has recently regained prominence as an alternative to Transformer-based architectures; its deterministic and interpretable methods appear promising for enabling logical reasoning (Chaudhuri et al., 2021). Interpretability is defined as a model's ability to accurately demonstrate its flow of reasoning to arrive at an answer. As the name suggests, it aims to combine the best of both worlds: neural networks and symbolic reasoning. The former enables learning, creativity, and inductive reasoning, while the latter handles logical reasoning with symbolic rules and algorithms.

However, many recent neurosymbolic works (Badreddine et al., 2022; Riegel et al., 2020) are not generalizable as evidenced by how they are typically benchmarked on domain-specific tasks. For example, the CLUTRR benchmark, which only includes family relations, is commonly used (Sinha et al., 2019) to evaluate inference skills. This is because such models require a comprehensive list of task-specific axioms/formulas to be defined before training; additional details and examples are provided in Section 2. Since it is unfeasible to manually define logical rules for hundreds of expansive topics, these models cannot be applied to domain-agnostic benchmarks like MMLU (Hendrycks et al., 2020), FOLIO (Han et al., 2022), and StrategyQA (Geva et al., 2021); they cannot achieve the broad applicability of existing LLMs. As such, we focus on general logical reasoning in this paper, which we define as the ability to deductively reason in domain-agnostic tasks.

Inspired by the taxonomies in Kautz (2020) and Ciatto et al. (2024), we identify two main approaches to neurosymbolic AI for general logical reasoning, i.e. integrative and hybrid, which are illustrated in Fig. 2. The integrative approach modifies the neural network architecture to allow it to perform logical reasoning in a deterministic and interpretable way. On the other hand, the hybrid approach sidesteps the limitations of traditional neural networks by coupling them with external symbolic solvers. Recent literature found success with both approaches for domain-specific benchmarks like CLUTRR and StepGame (Shi et al., 2022), but their use in general logical reasoning is still understudied. We elaborate further on this in Section 2.

To evaluate the merits of both approaches, we develop a best-in-class model for each approach and compare their strengths and weaknesses. For the integrative approach, we create a novel neural network that consists solely of differentiable logic gates, which we refer to as Logic Neural Network (LNN). It can deterministically represent any and all laws of propositional calculus. For example, the statement "If $a$ and $b$, then $c$ is true." can be represented by an AND logic gate ($a \wedge b$). Moreover, it is interpretable. Once a specific neuron has chosen a logic gate, one can precisely interpret the argument form used. Using a synthetic dataset, we experimentally validate that LNN's relaxation formula converges on the correct logic gate 3 times faster than Logic Gate Network (LGN) (Petersen et al., 2022), the existing state-of-the-art integrative model. Moreover, LNN outperforms LGN on the Breast Cancer dataset and performs comparably on the Adult Census dataset (Asuncion et al., 2007).

For the hybrid approach, we introduce LLM-SS, a framework that combines an LLM and a symbolic solver. It is, by design, model-agnostic, domain-agnostic, and requires little human input. Broadly speaking, in the case of question-answering (QA) tasks, the LLM is responsible for generating natural language premises for the question, and then translating them into logical form. Afterward, the logical form is fed to the symbolic solver, which outputs the final conclusion using deductive reasoning. LLM-SS achieved higher or similar performance and lower error rates on domain-agnostic QA tasks compared to other models using the hybrid approach through the use of several novel techniques.

To evaluate which approach holds more potential for general logical reasoning, we formulate our comparison based on the following criteria: (i) ability to reason symbolically, (ii) interpretability of reasoning chain, and (iii) retention of LLM abilities. We find that while both integrative and hybrid approaches are able to reason symbolically, the former's interpretability decreases when model size increases and the former loses much of the capabilities of existing LLMs, such as knowledge retrieval and generalization. LNN and the integrative approach as a whole suffer from theoretical limitations that limit their potential for tackling general logical reasoning. Given these factors, we contend that the hybrid approach is more promising. Finally, we propose a neurosymbolic framework based on LLM-SS to support future works in this direction.

## 2 Related Works

With regards to the integrative approach, Garcez & Lamb (2023) reviewed several systems where symbolic reasoning is contained within the neural network. Most notably, Logic Tensor Network (Badreddine et al., 2022) is designed to learn new predicates via a deep neural network while satisfying a first-order logic knowledge base. Daniele & Serafini (2019), Fischer et al. (2019), and Manhaeve et al. (2018) present models with similar principles. Logical Neural Network (Riegel et al., 2020), developed by IBM, creates a 1-on-1 correspondence between each neuron and a logic gate, a similar concept to our LNN.

However, the primary difference between the two aforementioned systems and LNN is that the former requires Real Logic axioms/formulas specific to the given task to be manually defined prior to training. The model subsequently learns the weights with respect to the axioms provided. LNN, on the other hand, consists of the same 6 logic gates for each neuron across any given task. It learns each neuron's optimal logic gate during training, thus dynamically constructing the best-performing axioms/formulas. This is a significant disadvantage for the Logic Tensor Network and Logical Neural Network because it limits their scope to problems with known, well-defined logical rules, thereby diminishing their usefulness in real-world applications. For example, IBM's Logical Neural Network is tested on the Lehigh University Benchmark (LUBM) (Guo et al., 2005), which contains predefined OWL axioms.

To the best of our knowledge, the only model architecture that allows for the choice of a logic gate is the Logic Gate Network (LGN) (Petersen et al., 2022). Its fundamental principle is the same as LNN: allow neural networks to choose between logic gates through differentiation by continuously relaxing them. However, LNN and LGN differ in how various logic gates are combined. This paper creates a distinct relaxation formula for each logic gate and then uses categorical probability distribution to create a weighted average across 16 logic gates. However, LGN's primary objective was to decrease inference time for computer vision tasks by using the highest probability logic gate for each neuron. Hence, its practical ability to converge to the appropriate logic gate for reasoning tasks remains unexamined.

On the other hand, the hybrid approach has seen state-of-the-art results on domain-specific reasoning tasks, but few have applied this approach to domain-agnostic tasks. Yang et al. (2023) applied GPT-3 and Clingo to the benchmark datasets: bABI, StepGame, CLUTRR, and gSCAN. However, the scope of its application is limited. For example, CLUTRR only involves the inference of family relationships. Moreover, the authors manually wrote an ASP knowledge module for each task, e.g. the one for CLUTRR contains an exhaustive list of family relationships. Therefore, Yang's architecture is not applicable to general QA tasks. Deepmind's AlphaGeometry (Trinh et al., 2024) combines an LLM with the symbolic engine DD+AR, which contains geometric rules. Similarly, its scope is limited to geometry problems. Moreover, these natural language geometry questions are manually translated into a domain-specific form in order for DD+AR to understand the problem. This is again unrealistic for general QA tasks. Models with similar frameworks, and therefore similar limitations include Silver et al. (2016), McGinness & Baumgartner (2024), Zhang et al. (2023), and Dai et al. (2019).

To our knowledge, only the Faithful-CoT model (Lyu et al., 2023) attempts to combine LLMs with symbolic engines for a wide range of tasks, without knowledge modules or manual translation of problems. However, using GPT-4 and Prolog, an alternative to Clingo, it achieved a mere 54% accuracy rate on the StrategyQA benchmark (Geva et al., 2021).

## 3 Model Architecture

We now explain the model architecture of our Logic Neural Network (LNN) and LLM-Symbolic Solver (LLM-SS), representing the integrative and hybrid approaches respectively.

### 3.1 Logic Neural Network (LNN)

LNN is a regular neural network with an adapted logic gate formula for each neuron, which builds upon the Logic Gate Network (Petersen et al., 2022). Every two neurons in a layer connect to a randomly chosen neuron in the subsequent layer. Each neuron has a choice of 16 distinct logic gates, as shown in Table 1. This is because, given 2 binary inputs, there are 4 unique input combinations. Since the output is also binary, there are 16 unique output combinations, each corresponding to a logic gate. Two neurons are connected instead of three or more because the latter's implementation is considerably more complicated, yet does not exhibit stronger performance empirically given a comparable number of parameters (Petersen et al., 2022; Benamira et al.).

Table 1: List of logic gates with their corresponding relaxation formulas and outputs values given input neurons $a$ and $b$. This table is derived from Petersen et al. (2022).

| Logic Gate | Real-Valued Logic | $A = 0, B = 0$ | $A = 1, B = 0$ | $A = 0, B = 1$ | $A = 1, B = 1$ |
|---|---|---|---|---|---|
| False | $0$ | 0 | 0 | 0 | 0 |
| $A \wedge B$ | $A \cdot B$ | 0 | 0 | 0 | 1 |
| $\neg(A \Rightarrow B)$ | $A - AB$ | 0 | 0 | 1 | 0 |
| $A$ | $A$ | 0 | 0 | 1 | 1 |
| $\neg(A \Leftarrow B)$ | $B - AB$ | 0 | 1 | 0 | 0 |
| $B$ | $B$ | 0 | 1 | 0 | 1 |
| $A \oplus B$ | $A + B - 2AB$ | 0 | 1 | 1 | 0 |
| $A \vee B$ | $A + B - AB$ | 0 | 1 | 1 | 1 |
| $\neg(A \vee B)$ | $1 - (A + B - AB)$ | 1 | 0 | 0 | 0 |
| $\neg(A \oplus B)$ | $1 - (A + B - 2AB)$ | 1 | 0 | 0 | 1 |
| $\neg B$ | $1 - B$ | 1 | 0 | 1 | 0 |
| $A \Leftarrow B$ | $1 - B + AB$ | 1 | 0 | 1 | 1 |
| $\neg A$ | $1 - A$ | 1 | 1 | 0 | 0 |
| $A \Rightarrow B$ | $1 - A + AB$ | 1 | 1 | 0 | 1 |
| $\neg(A \wedge B)$ | $1 - AB$ | 1 | 1 | 1 | 0 |
| True | $1$ | 1 | 1 | 1 | 1 |

However, logic gates are discrete, with values of either 0 or 1, and are therefore non-differentiable. Nonetheless, gradient descent is necessary for LNN to work. To achieve this, we first relax each discrete logic gate into a continuous graph, with 0 and 1 at each boundary, also known as real-valued logic. For example, the AND logic gate can be represented by $a \cdot b$ while the OR logic gate is represented by $a + b - a \cdot b$, based on probabilistic T-norm and T-conorm respectively (van Krieken et al., 2022). Notice that all formulas in Table 1 require at most inputs $a$, $b$, and $a \cdot b$, where $a$ and $b$ are the values of the input neurons. We can therefore further generalize the formulas into Eq. 1, where $a$ and $b$ are the input neurons, $c$ is the output neuron, and $w_{1-4}$ are trainable weights. The sigmoid function $\sigma$ ensures that output $c$ ranges from 0 to 1 during training. Each neuron there uses Eq. 1 during training.

$$c = \sigma(w_1 a + w_2 b + w_3 (a \cdot b) + w_4) \tag{1}$$

At inference time, Eq. 1 is then translated into a discrete logic gate for each neuron. Each output $o_i$ of the 4 input possibilities is discretized into 0 or 1, such that when $o_i > 0.5$, it is assigned a value of 1. Based on the 4 outputs, the corresponding logic gate is determined. All reported results are based on accuracies from the discretized version.

## 3.2 LLM-Symbolic Solver (LLM-SS)

We use a QA task to explain how LLM-SS works and to evaluate its performance. This is because QA tasks require substantial logical reasoning, while also having considerable emphasis on knowledge storage and retrieval, and inductive reasoning. This ensures the latter properties are not compromised in pursuit of logical reasoning.

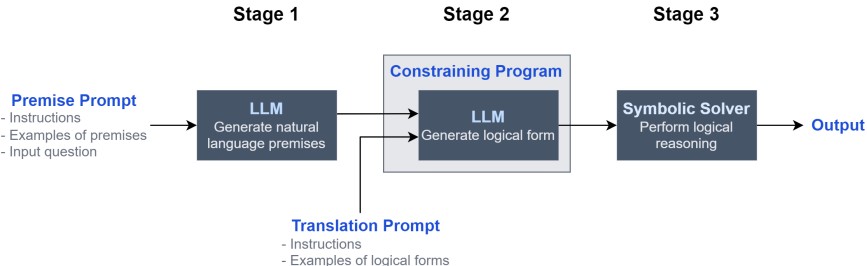

Figure 3: Architecture of LLM-SS model

The general structure of LLM-SS is illustrated in Fig. 3. It consists of three stages. In Stage 1, few-shot prompting is applied to a pre-trained LLM to generate natural language premises for an input question. Based on propositional logic, only two types of premises are allowed: (1) declarative sentences, which are statements that have truth values and no connectives, e.g. "A spider has 8 legs.", and (2) conditional sentences, which are essentially if-else statements, e.g. "If an animal has 6 legs, it is not a spider." (Pospesel, 1974). In other words, this is similar to conventional chain-of-thought (CoT) prompting (Wei et al., 2022), except the final answer is not generated and only two types of sentences are encouraged.

In the second stage, another few-shot prompt, together with the premises generated in Stage 1, is applied to a pre-trained LLM, which outputs the logical form of the given premises. Given that it translates natural language sentences into a machine-understandable representation, Stage 2 can be formulated as a semantic parsing task. However, given that LLM-SS must perform semantic parsing on uncontrolled natural language, which includes a wide range of vocabulary and grammatical structures, the examples provided in the prompt cannot cover the wide array of knowledge representations. This often leads to invalid logical forms, thus preventing the subsequent symbolic solver from producing an answer at all. In fact, Lyu et al. (2023) find that syntax errors and infinite loops (which we consider a subset of syntax errors) account for 52.9% of all errors made by OpenAI Codex on the StrategyQA dataset. To tackle this issue, we incorporate an LLM constraining program into Stage 2, which is a class of software that ensures the text generated by LLMs adheres to specific formats and rules (Beurer-Kellner et al., 2023). Its use for semantic parsing

tasks is currently underexplored in the literature. While the specific implementation varies, these programs generally work by identifying tokens that violate conditions defined by the user, in order to generate a token mask for the decode function. This ensures that the pre-trained LLM's choice of the next token is within the subset of valid tokens. For example, if an LLM constraining program follows a formalism that states that "=" must be followed by "True" or "False", then assuming the previous output is "=", the subsequent token must be either "True" or "False". We chose Microsoft Guidance (Microsoft, 2023) as our constraining program, which is open-source and actively supported.

In the third and final stage, a symbolic solver receives the logical forms generated in Stage 2 as input and then performs deductive reasoning on them to reach the final conclusion. It must be a deterministic model, where given a set of premises, it produces a logically entailed conclusion in all cases without fail; to ensure interpretability, the algorithms used throughout the process must also be transparent. To accomplish this, we turn to answer set programming (ASP) (Lifschitz, 2019), a form of logic programming (Lloyd, 2012). Logic programming represents natural language sentences as logical forms. Logical forms allow the logical relationships between entities to be unambiguously understood by the program, unlike natural language sentences by LLMs which may have multiple interpretations. ASP is a subset of logical programming, which focuses on solving search problems; finding the truth value of a statement based on a set of premises is one such problem. We use Clingo (Gebser et al., 2019) as our ASP solver due to its straightforward syntax and considerable open-source support. Below are examples of how natural language sentences, both declarative and conditional, can be represented in Clingo. Note the ability to use mathematical expressions in the last example.

- [Declarative] Sam has a cow.
  *no_of_cows_owned(sam, 1).*

- [Conditional] If Sam owns a cow, Sam is a farmer.
  *farmer(sam) :- no_of_cows_owned(sam, 1).*

- [Conditional] If Sam owns more than five cows, then he is rich.
  *rich(sam) :- no_of_cows_owned(sam, Number), Number >5.*

This explains the architectural choices in the earlier stages. Restricting the premises generated in Stage 1 to declarative and conditional forms allows a simpler and therefore more error-free translation into logical form; using LLM constraining software in Stage 2 helps ensure the LLM adheres to Clingo's strict syntax rules.

Importantly, so long as the premises and ASP code are accurate, the final output is necessarily true due to the deterministic nature of Clingo. This is unlike a traditional chain-of-thought (CoT) model (Wei et al., 2022), where the final output is also generated by the LLM. A CoT model's final output may not be consistent with its premises. For example, suppose we ask: "Are all the elements plants need for photosynthesis present in Mars' atmosphere?" (derived from the StrategyQA dataset), the output may be as follows:

1. Plants need three elements for photosynthesis: Hydrogen, Oxygen, and Carbon.

2. The atmosphere of Mars is composed of carbon dioxide, nitrogen, argon, and trace levels of water vapor, oxygen, carbon monoxide, hydrogen, and other noble gases.

3. Therefore, not all the elements plants need for photosynthesis are present in Mars' atmosphere.

The conclusion does not follow from the premises, which is endemic to the lack of logical reasoning in traditional chain-of-thought systems. By transferring the responsibility of logical reasoning to an ASP like Clingo, this issue is completely eliminated.

## 4 Experimental Setup

We now benchmark LNN and LLM-SS against other methods with integrative and hybrid approaches respectively, to test whether they are, in fact, best-in-class models.

### 4.1 Logic Neural Network (LNN)

Our experiments for LNN aim to understand whether (i) the neurons are able to converge on the appropriate logic gate, and (ii) its performance on existing benchmarks. Both objectives are empirically studied in Experiments 1 and 2 respectively. This will help us understand the integrative approach's pros and cons, which we discuss in Section 6.1

For **Experiment 1**, we create a simple synthetic dataset where the model must identify the appropriate logic gate when provided with 4 unique sets of neuron $a$ and $b$ values and their corresponding output. For example, with reference to Table 1, when provided the outputs 0, 0, 0, 1 for their corresponding $a$ and $b$ values, the model is expected to identify the $A \wedge B$ logic gate. We benchmark LNN against the state-of-the-art Logic Gate Network (LGN) model (Petersen et al., 2022) based on (i) whether it converges on the correct logic gate and (ii) how many iterations it takes. Models like Logical Neural Networks and Logic Tensor Networks are excluded because they presuppose that the model already knows the logic gates used before training. This condition does not hold for the aforementioned task, thus making it beyond these models' existing capabilities.

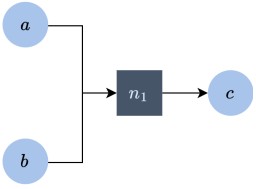

Figure 4: Experiment 1: Model Architecture of LNN and LGN

The architecture of both models is shown in Figure 4. It is a single-layer model, containing just 1 neuron with the respective formulas of LNN and LGN. When the neuron's discretized version chooses the appropriate logic gate, it is considered to have converged. As for hyperparameters, both models are trained with 1000 iterations, a learning rate of 0.01, and using the Adam optimizer (Kingma & Ba, 2015).

For **Experiment 2**, we utilize the Adult Census and Breast Cancer datasets (Asuncion et al., 2007), which are classification tasks containing 48842 and 286 instances respectively. LNN is benchmarked against LGN and a multi-layer perception (MLP). The model architecture and hyperparameters of LGN and MLP are determined by the recommended settings in the original paper Petersen et al. (2022), which are as shown in Table 2. Importantly, model sizes are kept roughly equivalent in the interest of fairness. All models are trained up to 200 epochs at a batch size of 100. The MLPs are ReLU activated.

| Model | Space | Layers | Neurons per layer |
|---|---|---|---|
| **Breast Cancer** | | | |
| Logic Neural Network | 320B | 5 | 128 |
| Logic Gate Network | 320B | 5 | 128 |
| Multi-Layer Perceptron | 1.4KB | 2 | 8 |
| **Adult** | | | |
| Logic Neural Network | 640B | 5 | 256 |
| Logic Gate Network | 640B | 5 | 256 |
| Multi-Layer Perceptron | 15KB | 2 | 32 |

Table 2: Model architecture and hyperparameters for LNN's Experiment 2

### 4.2 LLM-Symbolic Solver (LLM-SS)

The task chosen is the StrategyQA dataset (Geva et al., 2021), where the model must infer the appropriate premises based on the question and reason about those premises. Importantly, it prevents LLM-SS from being biased towards any domain because the dataset may contain questions about any topic at all, including anything from historical knowledge to chemistry. Fig. 5 shows how LLM-SS will answer the question "Was Jackson Pollock trained by Leonardo da Vinci?". The model must first infer in stage 1 that the question can be answered by the years in which they were alive, i.e. 1912-56 for Pollock and 1452-1519 for Da Vinci. It must then reason that since these years do not overlap, one could not have trained the other. In order to ensure that the conclusion follows from the premises, Stage 2 converts the premises above into Clingo

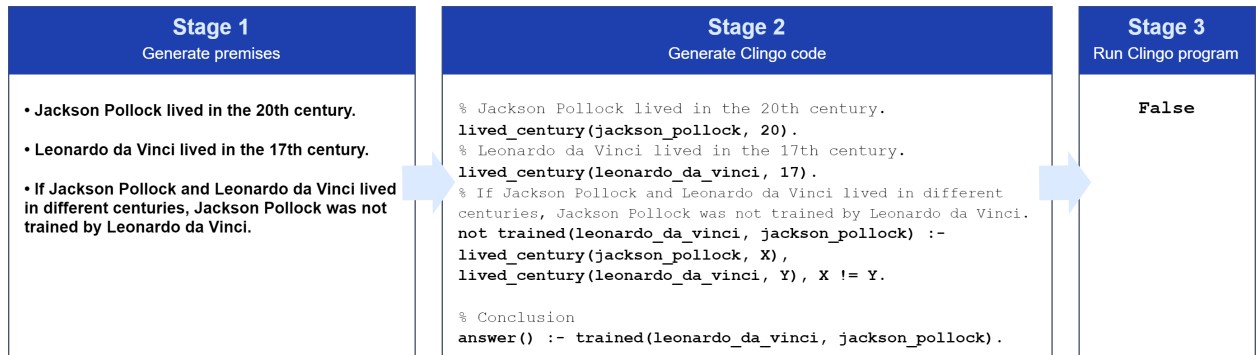

Figure 5: Example response by LLM-SS to the question "Was Jackson Pollock trained by Leonardo da Vinci?"

code, i.e. a logical form, which is then deterministically run on the Clingo program in Stage 3 to generate the final answer. This benchmark fits our purpose because, beyond testing for general logical reasoning, it also requires the advantages of traditional neural networks: (i) learning, storing, and retrieval of knowledge, which is evaluated by whether models have the facts needed for the question from their training data, and (ii) inductive reasoning, which is evaluated by whether models can select the relevant facts.

We compare LLM-SS to a traditional CoT model using (i) an LLM, (ii) an unconstrained LLM-SS, where the constraining software in Stage 2 is removed, and (iii) Faithful-CoT. LLM-SS uses Llama2-7B (Touvron et al., 2023) in Stage 1 and CodeQwen1.5-7B (Bai et al., 2023) in Stage 2; unconstrained LLM-SS and the traditional CoT model uses Llama2-7B; Faithful-CoT uses GPT-4. Unconstrained LLM-SS combines Stages 1 and 2, since the insertion of natural language premises as code comments are done via the constraining software, thus only one LLM is used. This aligns with the approach of Faithful-CoT. All models use Clingo as their symbolic solver, except Faithful-CoT, which uses Prolog, an alternative logic programming language. Few-shot prompts are executed with four examples only, besides Faithful-CoT, which uses six. We use fewer examples because LLM-SS uses Llama2-7B, which has a smaller context length than GPT-4. This reduces the probability of exceeding the maximum context length. As for metrics, other than accuracy, we also measure the error rate, which is the percentage of questions with no answers generated. This happens when the CoT model does not produce a "yes" or "no" answer, or when the ASP code has a syntax error.

## 5 Results

### 5.1 Logic Neural Network (LNN)

The results of **Experiment 1** are shown in Table 3. LNN's accuracy of 100% proves that Eq. 1 is able to converge on the correct logic gate for all 16 options. Moreover, while LGN also achieves 100% accuracy, it converges almost 3 times slower than LNN, suggesting the latter's relaxation formula may be more optimal for training.

Table 3: Results of Experiment 1

|  | Accuracy | Avg. Iterations Needed |
|---|---|---|
| Logic Gate Network | 1.00 | 163.4 |
| Logic Neural Network | **1.00** | **63.9** |

As for **Experiment 2**, the results are shown in Table 4. LNN achieves the highest accuracy (78.6%) out of the 3 models for the Breast Cancer dataset. The Adult Census, on the other hand, saw comparable results for all models, with MLP being marginally better (84.9%) than the rest. These results suggest that LNN outperforms LGN on smaller datasets due to the former's stronger convergence abilities. However, as

datasets increase in size, LGN starts to outperform LNN. While further research is required to understand and optimize LNN's performance and behaviors on larger problems, this is not necessary for our analysis in Section 6.1 and therefore beyond the scope of our work.

Table 4: Results of Experiment 2

|  | Breast Cancer | | Adult | |
| --- | --- | --- | --- | --- |
|  | Accuracy | Space | Accuracy | Space |
| Logic Neural Network | **0.786** | 320B | 0.847 | 640B |
| Logic Gate Network | 0.761 | 320B | 0.848 | 640B |
| Multi-Layer Perceptron | 0.753 | 1.4KB | **0.849** | 15KB |

## 5.2 LLM-Symbolic Solver (LLM-SS)

As shown in Table 5, LLM-SS has a significantly higher accuracy and lower error rate compared to its unconstrained counterpart. It is evident that constraining LLM generation to enforce the syntax of Clingo leads to fewer errors during code execution, thereby increasing LLM-SS's accuracy. This is further highlighted when benchmarked against Faithful-CoT, which suggests a smaller constrained model, e.g. Llama2-7B, can perform comparably to a larger unconstrained model, e.g. GPT-4. However, LLM-SS still lags behind the traditional CoT model in terms of accuracy.

Table 5: Results of LLM-SS experiment

|  | Accuracy | Error Rate (%) |
| --- | --- | --- |
| CoT | 60.6 | 0.6 |
| Faithful-CoT | 54.0 | - |
| LLM-SS (Unconstrained) | 48.5 | 17.8 |
| **LLM-SS** | 54.0 | 1.5 |

The main bottleneck of LLM-SS's accuracy can be straightforwardly deduced. The process in which the CoT model gathers facts is identical to Stage 1 of LLM-SS. Stage 3 of LLM-SS is a deterministic execution of Clingo, so it cannot be blamed for any errors. Thus, Stages 1 and 3 cannot explain the gap in accuracy between the two models. Thus, Stage 2 is the cause, specifically, the translation from natural language sentences to code. McGinness & Baumgartner (2024) categorizes translation errors into syntactic and semantic errors. Syntactic errors are defined as errors in the logical form that prevent parsing, while semantic errors are defined as logical forms that falsely represent their corresponding sentence despite being parsable. Given that the syntactic error rate for LLM-SS is 2.5%, it is evident that semantic errors are mostly responsible. Semantic error manifests in several ways: First, the naming convention between premises is sometimes inconsistent. For example, one premise may say "1519" while another may say "16th century", thus making the use of math operators to compare between them impossible. Second, the code translation may also be nonsensical, such as the usage of words and phrases that do not even appear in the premise.

## 6 Discussion

### 6.1 Comparison of Integrative & Hybrid Approaches

Is the integrative or hybrid approach more promising for developing general logical reasoning without sacrificing the capabilities of existing LLMs? Given the strong performance of LNN and LLM-SS against state-of-the-art models using either approach, they serve as case studies to answer the aforementioned question. We compare the approaches using the following criteria: (i) ability to reason symbolically, (ii) interpretability of reasoning chain, and (ii) retention of LLM abilities, i.e. whether the model can still preserve the advantages of existing LLMs, such as memorization and generalization. Table 6 summarizes our analysis.

Table 6: Comparison of integrative and hybrid approaches

| | Symbolic Reasoning | Interpretability | LLM Abilities |
|---|---|---|---|
| Integrative | ✓ | ∼ | ✗ |
| Hybrid | ✓ | ✓ | ✓ |

**Criteria 1: Symbolic Reasoning.** LNN is able to logically reason, albeit in a limited fashion. Specifically, it is restricted by the connections pre-formed between neurons of one layer to the next. Realistically, pre-formed connections may not be the most accurate representation of a given logical argument; models should be allowed to learn the most optimal connections. Gumbel-Max Equation Learner Networks (Chen, 2020) uses Gumbel-Softmax to learn which outputs of the previous layer should be the input of the next layer. However, each layer's arithmetic operations are predefined. Since combining both flexible logical formulas and flexible connections has not been achieved but appears plausible, we argue that the integrative approach is capable of symbolic reasoning.

The hybrid approach, on the other hand, is capable of symbolic reasoning due to the use of symbolic programs. While Stage 2 currently hinders reasoning due to its inadequate translation abilities, this can be addressed by designing improved natural language-to-Clingo code translation models. For future development, inspiration can be drawn from semantic parsing tasks like Abstract Meaning Representation (AMR) (Knight et al., 2021), where a sentence's meaning is parsed into a tree structure. Effective methods for AMR parsing include structure-aware transition-based approaches with pre-trained language models (Zhou et al., 2021) and model ensembling (Lee et al., 2022).

**Criteria 2: Interpretability.** Both approaches contain some degree of interpretability given their explicit use of symbolic reasoning: the logic gates used by LNN and the Clingo code used by LLM-SS are readily accessible. However, the former's interpretability scales poorly. Consider the LNN implemented for the Breast Cancer dataset which contains $5 \times 128 = 640$ neurons. While one can identify the logic gate of each neuron, a human cannot interpret a line of reasoning containing 640 logic gates. Therefore, LNN is only realistically interpretable when restricted to just a few logic gates. From a human perspective, as the number of neurons increases, LNN's reasoning ability becomes no different from a traditional neural network.

For the hybrid approach, Lyu et al. (2023) point out that while the Problem-Solving stage in Faithful-CoT, which corresponds to Stage 3 in LLM-SS, is transparent and interpretable, the Translation Stage, which corresponds to Stages 1 and 2 in LLM-SS, is still opaque. In other words, one cannot interpret how premises and logical form translations are generated. However, we argue that this is not a fundamental issue. Consider an analogy to the human brain: when humans make an argument, it is deemed logical so long as the premises retrieved from memory lead to a conclusion. The criteria for an interpretable line of reasoning do not require transparency into how our brain retrieved the premises in the first place. Similarly, for LLM-SS, Stages 1 and 2 do not necessarily need to be deterministic and interpretable for logical reasoning to occur.

**Criteria 3: Retention of LLM Abilities.** The Transformers architecture, with its maze of linear and non-linear transformations, excels at inductive reasoning, and the learning, storage, and retrieval of knowledge. To compete against existing LLMs in general logical reasoning, neurosymbolic models must retain the above capabilities, otherwise they will be relegated to solving minute, small-scale problems. The integrative approach replaces the Transformers architecture with a logic-based one, thereby entirely stripping the model of its LLM capabilities. Moreover, attempting to attain the advantages of LLMs conflicts with the reasoning capabilities of integrative models. For example, when LNN is scaled to hundreds or thousands of neurons, it is no longer interpretable.

The hybrid approach, on the other hand, has no such issues. In LLM-SS, Stage 1 still uses a Transformer-based architecture, i.e. Llama2-7B, thereby retaining the capabilities of existing LLMs, while Stage 3 uses Clingo to perform symbolic reasoning. This separation of responsibilities allows LLM-SS to achieve the best of both worlds.

To summarize, the hybrid approach is superior to the integrative approach with regard to interpretability and retention of LLM abilities. We therefore posit that the hybrid approach is more promising for developing general logical reasoning and we encourage further research in this direction.

## 6.2 LLM-SS: A framework for hybrid approach

Importantly, the framework exemplified by LLM-SS in Fig. 3 is generalizable for future neurosymbolic efforts. By segmenting LLM-SS into three distinct stages, it becomes a modular framework, where each stage can be independently tested, updated, and improved. Specifically, the choice of models for each stage is also flexible: alternative pre-trained LLMs like GPT-4 and Gemini 1.5 can be used in Stage 1, semantic parsing models can be used in Stage 2, and alternative symbolic solvers like Datalog and Python can be used in Stage 3.

Most neurosymbolic models are tailored to solve domain-specific tasks by using domain-specific knowledge modules and manually translated logical forms. In contrast, LLM-SS is designed for general logical reasoning, ensuring its viability across a variety of tasks, regardless of domain. To apply this framework to a new task, only in-context learning is required as human input.

To improve this neurosymbolic framework for general logical reasoning, further exploration may include the (i) optimization of prompts and choice of models, (ii) mitigation of semantic errors in Stage 2, (iii) use of alternative logical forms in Stage 2 and 3, such as knowledge graphs (Pan et al., 2024), and (iv) evaluation of LLM-SS's effectiveness in other domain-agnostic tasks.

## 7 Conclusion

Our study finds that the hybrid approach to neurosymbolic AI is more promising than the integrative approach for developing general logical reasoning without sacrificing the capabilities of existing LLMs. To aid our analysis, we introduced the Logic Neural Network (LNN) and LLM-Symbolic Solver (LLM-SS) which serve as case studies and representatives for the integrative and hybrid approaches respectively. Finally, to support future research in neurosymbolic AI, we propose LLM-SS as a modular, model-agnostic, and domain-agnostic framework.

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

## A   Appendix

You may include other additional sections here.

