# OpenReview forum: "What else does attention need: Neurosymbolic approaches to general logical reasoning in LLMs?"
_TMLR — Withdrawn by Authors_

### Review · Reviewer_z3YB · 2024-11-12

**Summary Of Contributions:**

This paper identifies and investigates two neurosymbolic approaches for enhancing general logical reasoning: the integrative approach, where logical reasoning is contained in the neural network architecture, and the hybrid approach, which combines neural models with an external symbolic solver. The authors introduce the Logic Neural Network (LNN) and the LLM-Symbolic Solver (LLM-SS) as representative models of these two approaches, respectively, and benchmark their performance on domain-agnostic reasoning tasks. Through experiments and analysis, the paper demonstrates that the hybrid approach is more promising for developing general logical reasoning, as it achieves higher interpretability while preserving the capabilities and strengths of existing LLMs. To facilitate further research, the authors propose LLM-SS as a modular, flexible, and domain-agnostic framework.

**Audience:**

Yes

**Broader Impact Concerns:**

There may be ethical concerns related to the reliance on LLMs, and it would be better to include a Broader Impact Statement.

**Claims And Evidence:**

Yes

**Requested Changes:**

1) Provide a direct comparison or benchmark between the integrative and hybrid approaches.
2) Clarify the differences between LLM-SS and Faithful-CoT and its novelty, and improve the performance to achieve better results. (critical)
3) Expand the experimental evaluation to include varied hyperparameter settings and additional datasets.
4) Analyze the failed cases of LLM-SS in order to identify and address the specific stages or components contributing to the 46% failure rate.

**Strengths And Weaknesses:**

**Strengths.**

1) The paper provides a detailed comparison between the integrative and hybrid neurosymbolic approaches, analyzing both quantitative performance metrics and qualitative aspects such as interpretability. The analysis shows that the hybrid approach is more promising for general logical reasoning, as it offers greater interpretability while preserving the capabilities of large language models (LLMs).

2) The authors introduce the Logic Neural Network (LNN) and the LLM-Symbolic Solver (LLM-SS). LNN, an adaptation of the Logic Gate Network (LGN), demonstrates faster convergence as reported in the experiments. LLM-SS integrates symbolic deduction with LLMs, offering a modular framework for reasoning tasks.


**Weaknesses**

1) The paper lacks a direct comparison between the integrative and hybrid approaches. While such a comparison might be challenging, developing a benchmark to facilitate this would strengthen the argument that the hybrid approach is more promising.

2) The construction of LLM-SS lacks sufficient novelty. It is unclear how LLM-SS meaningfully differs from Faithful-CoT. Additionally, the design of LLM-SS closely resembles existing neurosymbolic solvers and autoformalization processes in theorem proving. For example, Stages 1 and 2 in LLM-SS are similar to the process of generating formal statements or theorems. However, the generated formal premises/claims/rules by LLMs are neither verified nor guaranteed to be accurate, which can result in errors. Furthermore, the generated statements may not be in the desired form, preventing the symbolic solver from producing a valid answer.

3) The experiments involving LNN, LGN, and MLP are limited in scope. It is not reported how changes in hyperparameters, such as those in MLP, might affect the results. Expanding the experimental setup to include a broader range of configurations and datasets would provide a clearer understanding of the models’ relative strengths and weaknesses.

4) The performance of both LNN and LLM-SS is underwhelming. In the experiments, MLP outperforms LNN and LGN in one task, and their overall accuracy is similar. For LLM-SS, the CoT model achieves higher accuracy and lower error rates and Faithful-CoT matches the performance of LLM-SS. These results cast doubt on whether the proposed models offer practical improvements over existing approaches.

5) The paper does not provide an analysis of the failed examples of LLM-SS. With multiple stages involved in the reasoning process, it is unclear why the model failed on 46% of the test set. A detailed discussion of these failures would provide valuable insights and help guide future improvements.

---

### Review · Reviewer_2DUQ · 2024-11-15

**Summary Of Contributions:**

This paper investigates two main approaches to enhance the logical reasoning abilities of LLMs, the integrative approach and the hybrid approach. The paper proposes two new methods, LNN and LLM-Symbolic Solver, based on the two approaches, respectively. Then, the paper compares LNN and LLM-Symbolic Solver with previous methods in terms of accuracy and error rate.

**Audience:**

Yes

**Claims And Evidence:**

Yes

**Requested Changes:**

1.	The related work section mentions that the paper “uses categorical probability distribution to create a weighted average across 16 logic gates.” What does “weighted average” mean here? It seems that later parts of the paper does not mention it.

2.	Please see “weaknesses” for other questions.

**Strengths And Weaknesses:**

Strengths:

1.	The motivation is clear. The paper focuses on problems with LLM reasoning that conclusions may not result from premises.

2.	The paper is easy to read and follow.

Weaknesses:

1.	The use of the words “deterministic” and “probabilistic” seem confusing to me. My first thought is that “deterministic” refers to a standard neural network while “probabilistic” refers to a Bayesian neural network whose parameters are represented as probabilistic distributions. However, it is not the case in this paper.

2.	In Table 1, the columns “A=1,B=0” and “A=0,B=1” need to be exchanged.

3.	About the continuous relaxation of the logical gates in Eq. (1). Since the weights $w_1$ to $w_4$ can be arbitrarily learned, the function in Eq. (1) does not necessarily behave like a logical gate. Although in the inference stage, the output of the function is intentionally discretized to have binary values, the function itself can be far from a logical gate in the training stage.

4.	The **Experiment 1** for LNN is too simple as it only tests a one-neuron model. For such a simple model, I would encourage the authors to conduct some theoretical analysis on the convergence rate in addition to pure empirical observations. On the other hand, the experiment itself should focus on deeper architectures (at least two layers) in order to convince readers that LLN converges faster than the previous LGN method.

5.	In the **Experiment 2** for LNN, the paper uses a 2-layer MLP as a baseline method. The paper states that the reason for this choice is make “model sizes roughly equivalent in the interest of fairness.” This is somehow tricky: (1) The width of the MLP is not reported in the paper. So why not try a deeper MLP with less neurons in each layer? (2) The sizes of these networks are already very small (1-10KB), so I doubt whether controlling the parameter size is a reasonable setting.

6.	In Table 5, Faithful-CoT significantly underperforms traditional CoT. However, in the original paper of Faithful-CoT, it beats traditional CoT on nine out of ten benchmarks. I encourage the authors to explain why the reported results are quite different from previous works.

7.	The experiment on LLM-SS is only conducted on one dataset (StrategyQA), which is not sufficient to validate the effectiveness of LLM-SS. More tasks/benchmarks should be included.

8.	The proposed methods LLN and LLM-SS do not attain a reasonable performance gain over previous methods. Although I appreciate the analysis of why LLM-SS underperforms traditional CoT, but the takeaway from this analysis is still not clear.

---

### Review · Reviewer_sqPD · 2024-12-05

**Summary Of Contributions:**

To address LLMs' limitations in logical reasoning, including lack of determinism and interpretability, this paper distinguishes between integrative and hybrid neurosymbolic approaches. The author studies both approaches by training LNN and LLM-SS, and highlights that the hybrid approach retains LLM capabilities and better addresses general logical reasoning tasks.

**Audience:**

Yes

**Claims And Evidence:**

Yes

**Requested Changes:**

1. As mentioned in the Weakness, the overall performance needs to be improved.

2. More experimental details should be added to the main paper or appendix. Please also clarify the dataset or benchmark used in each result.

3. The authors could rethink the relationship between the two approaches. The conclusion is less rigorous.

**Strengths And Weaknesses:**

Strength:
* The paper makes a good point of defining and comparing integrative and hybrid neurosymbolic approaches.
* The paper is well-written and clear. The reasoning chain is explicitly represented in a symbolic solver, which is interesting.

Weakness:
* The authors claim the hybrid approach is better than the integrative one. I don't think this is a fair comparison since they are not using the same base model or dataset/benchmark.
* The proposed method, LLM-SS, achieves lower accuracy on some benchmarks compared to the vanilla chain-of-thought (CoT) models. The performance needs to be improved.
* The experiment section is limited. LLM-SS is only evaluated on one dataset. More benchmarks like MMLU are only mentioned in the introduction but not touched on in the experiments.

---

### Note · Authors · 2024-12-13

**Comment:**

We would like to thank the reviewers for their constructive feedback. The reviewers have suggested increasing the number and depth of experiments and improving the model performance. These changes cannot be adequately implemented within the discussion period, so we have decided to withdraw our manuscript. Once again, thank you for your valuable comments.

**Withdrawal Confirmation:**

I have read and agree with the venue's withdrawal policy on behalf of myself and my co-authors.